# Boundary-aware Contrastive Learning for Semi-supervised Nuclei Instance Segmentation

**Ye Zhang**[1]                     ZHANGYE94@STU.HIT.EDU.CN
**Ziyue Wang**[1]                    200111326@STU.HIT.EDU.CN
**Yifeng Wang**[2]                   WANGYIFENG@STU.HIT.EDU.CN
**Hao Bian**[3]                     H2495067728@GMAIL.COM
**Linghan Cai**[1]                    CEILINGHANS@GMAIL.COM
**Hengrui Li**[4]                    23B903028@STU.HIT.EDU.CN
**Lingbo Zhang**[3]              ZHANG-LB23@MAILS.TSINGHUA.EDU.CN
**Yongbing Zhang**[1,✉]                  YBZHANG@.HIT.EDU.CN

[1] *School of Computer Science and Technology, Harbin Institute of Technology, 518055, China.*

[2] *School of Science, Harbin Institute of Technology, 518055, China.*

[3] *Tsinghua Shenzhen International Graduate School, Tsinghua University, 518071, China.*

[4] *Faculty of Computing, Harbin Institute of Technology, 150001, China.*

**Editors:** Accepted for publication at MIDL 2024

## Abstract

Semi-supervised segmentation methods have demonstrated promising results in natural scenarios, providing a solution to reduce dependency on manual annotation. However, these methods face significant challenges when directly applied to pathological images due to the subtle color differences between nuclei and tissues, as well as the significant morphological variations among nuclei. Consequently, the generated pseudo-labels often contain much noise, especially at the nuclei boundaries. To address the above problem, this paper proposes a boundary-aware contrastive learning network to denoise the boundary noise in a semi-supervised nuclei segmentation task. The model has two key designs: a low-resolution denoising (LRD) module and a cross-RoI contrastive learning (CRC) module. The LRD improves the smoothness of the nuclei boundary by pseudo-labels denoising, and the CRC enhances the discrimination between foreground and background by boundary feature contrastive learning. We conduct extensive experiments to demonstrate the superiority of our proposed method over existing semi-supervised instance segmentation methods.

**Keywords:** Semi-supervised learning, Nuclei instance segmentation, Edge denoising.

## 1. Introduction

Nuclei instance segmentation is essential in the quantitative analysis of pathological images. The characteristics of nuclei, including their size, morphology, and distribution, can provide valuable insights into the tumor microenvironment, thereby offering crucial support for cancer diagnosis, staging, and grading processes (Khened et al., 2021; Hollandi et al., 2022). In recent years, deep learning techniques have made remarkable advancements in nuclei segmentation(Zhang et al., 2024a,b). DCAN (Chen et al., 2016) adopts a dual-branch decoder architecture to predict semantics and contours simultaneously to enhance the instance distinguishing. HoverNet (Graham et al., 2019) incorporates distance and gradient constraints

ZHANG WANG[1] WANG[2] BIAN[3] CAI[1] LI[4] ZHANG[3] ZHANG[1,✉]

to split individual instances effectively. Similar methods such as CDNet (He et al., 2021), and CellPose (Stringer et al., 2021) are also proposed to address overlapping nuclei challenges. However, these supervised methods typically rely on pixel-level annotations, which are time-consuming and labor-intensive and need professional guidance, hindering the development of models. Therefore, developing a technique that can effectively address the dependency on manual annotation for nuclei instance segmentation is crucial.

A common approach to address the problem of scarce labeled data is semi-supervised learning (Reddy et al., 2018; Van Engelen and Hoos, 2020). During the training process, abundant unlabeled and insufficient labeled data are used to train the network. The existing semi-supervised methods mainly leverage prior information to improve the pseudo-label quality. For example, ShapeProp (Zhou et al., 2020b) combines the information from bounding boxes and partially annotated masks to improve the segmentation accuracy of target regions based on Mask R-CNN (He et al., 2017). PAIS (Hu et al., 2023) uses a dynamic alignment loss to address the misalignment problem between classification and segmentation results, and then a new threshold filtering method for pseudo-labels is proposed. PointWS-SIS (Kim et al., 2023) balances false negative and false positive errors by utilizing point supervision prior information. However, due to the low color contrast differences between the nuclei and tissues, these methods still have defects in generating nuclear pseudo-labels, limiting the application of semi-supervised instance segmentation in pathological images.

Some methods use pseudo-label optimization strategies to enhance nuclei segmentation accuracy in semi-supervised scenarios. MMT-PSM (Zhou et al., 2020a) integrates multiple data-augmented segmentation results to construct reliable predictions and enhance pseudo-labels' confidence. CDCL (Wu et al., 2022) uses feature contrastive learning to promote feature consistency between the teacher and student networks, thus improving the quality of pseudo-labels. PG-FANet (Jin et al., 2024) employs a pseudo-label guided module that aggregates multi-scale, multi-stage features to enhance segmentation performance. However, nuclei exhibit diversity in morphology and size, and in cases with limited annotations, it is challenging for the teacher network to capture the complete range of nuclei shape features. Consequently, the generated pseudo-labels often contain edge noise because existing pseudo-label optimization methods lack specific designs for denoising nuclei boundaries, which always leads to inaccurate nuclei boundary predictions.

In this paper, to address the issue of boundary noise in nuclei segmentation, we propose a coarse-to-fine **b**oundary-**a**ware contrastive learning network for **s**emi-supervised nuclei **s**egmentation (BASS[1]). Firstly, we design a low-resolution denoising (LRD) segmentation head that promotes boundary smoothness. Additionally, within this segmentation head, we use a low-weight loss for the nuclei boundary region optimization, which reduces the impact of uncertain boundary prediction during training. Secondly, to minimize boundary noise further, we design a cross-RoI contrastive learning (CRC) module that finely partitions the internal, external, and boundary regions of nuclei, enhancing the discriminative capability of nuclei boundary features. To demonstrate the effectiveness of our proposed method, we conduct comparative experiments and ablation studies on two public datasets. The experimental results show that our proposed method outperforms existing semi-supervised methods, and the ablation studies demonstrate the effectiveness of the proposed modules.

---

1. Our code is availiable at https://github.com/zhangye-zoe/BASS.

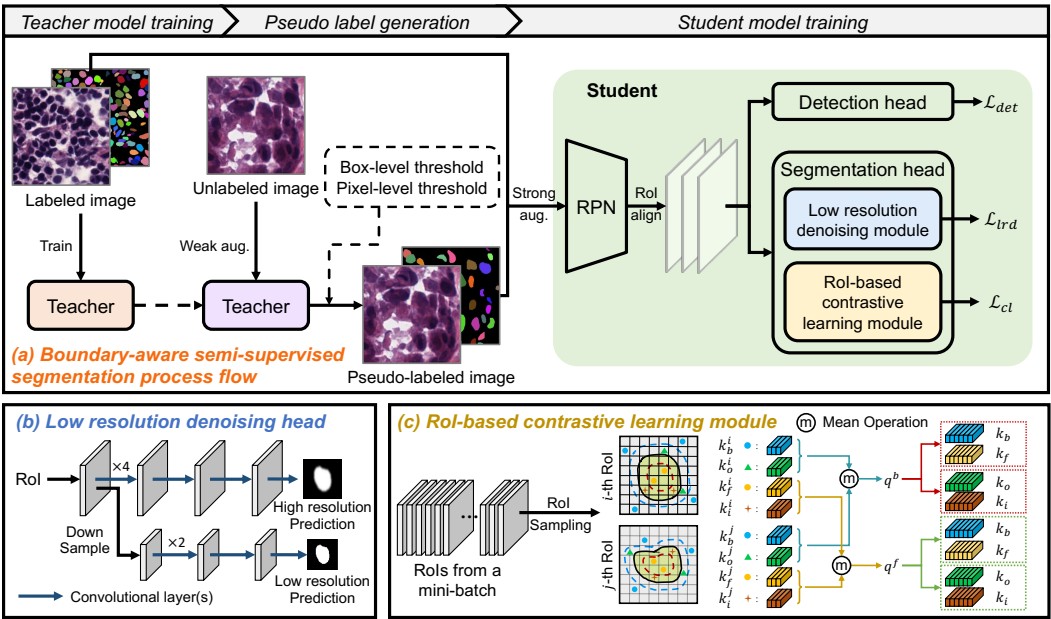

Figure 1: The framework of our semi-supervised nuclei segmentation method. (a)The training flow of our BASS. First, the teacher model generates pseudo-labels, and then the student model is used to train the nuclei segmentation network. (b) and (c) is the proposed low-resolution denoising module and cross-RoI contrastive learning module.

## 2. Methodology

### 2.1. Framework Overview

To address the boundary noise problem of nuclei segmentation under a semi-supervised scenario, we propose a coarse-to-fine boundary-aware denoising model, as shown in Fig. 1. Our whole training process can be divided into three stages. First, the labeled data $D_L = \{(x_i, y_i)\}_{i=1}^N$ is used to train a teacher model. In this step, we employ Mask R-CNN (He et al., 2017) as the baseline, and the loss function of the teacher network is defined as follows:

$$Loss^t = L_{seg}^t + L_{det}^t, \tag{1}$$

where $L_{seg}^t$ is the loss of the segmentation head, and $L_{det}^t$ is the loss of the detection head, which consists of the classification loss and regression loss. Then, the trained teacher network is employed to generate pseudo-labels $y_j^p$ for input $x_j$. To reduce the uncertainty of pseudo-labels, we employ box and pixel threshold filtering to generate high-confidence pseudo-labels. Finally, we combine the labeled data $D_L$ and the generated pseudo-labeled data $D_U = \{(x_j, y_j^p)\}_{j=1}^M$ to train the student network.

In the student network, our developed denoising methods are shown in the green box of Fig.1, which consists of a low-resolution denoising (LRD) module and a cross-RoI contrastive learning (CRC) module. The LRD employs low-resolution pseudo-labels as supervision information to promote the smoothness of nuclei contours. Meanwhile, the CRC utilizes boundary-aware contrastive learning to enhance the discriminative capability of contour

ZHANG WANG[1] WANG[2] BIAN[3] CAI[1] LI[4] ZHANG[3] ZHANG[1,✉]

features. In the training process, the overall loss is designed as follows:

$$Loss^s = L_{det}^s + L_{nmh} + L_{lrd} + L_{cl}, \tag{2}$$

where $L_{nmh}$ represents the naive high resolution segmentation loss, $L_{lrd}$ represents the low-resolution segmentation loss, and $L_{cl}$ represents the contrastive learning loss.

## 2.2. Threshold Filtering

Before training the student network, we chose high-confidence instances as pseudo-labels to reduce the uncertainty of the samples during student network training. In the pseudo-label generation stage, the teacher network outputs probability values for instances (box value) and mask probabilities (pixel value). We consider the pixel-level threshold $v_p$ to be the hyperparameter. As for the box-level threshold $v_b$, we assume the nuclear number distribution is consistent between labeled and unlabeled data. Based on this assumption, we uniformly sample 91 values between 0.1 and 1.0. Then, we iteratively apply these probabilities to filter the instances and calculate the number distribution of nuclei in the unlabeled data. Finally, we select the threshold closest to the distribution of labeled data. We validate the effectiveness of the threshold filtering method in ablation experiments.

## 2.3. Low-resolution Denoising Module

In the naive Mask R-CNN (He et al., 2017), the RoI head outputs a $14 \times 14$ feature map containing boundary noise. In the subsequent convolution process, Mask R-CNN increases the size of the feature map to capture more semantic information, but the boundary noise is also amplified. To avoid amplified noise effects, we design a low-resolution denoising module as shown in Fig.1(b), which utilizes the low-resolution pseudo-labels as supervision for model training. In the LRD, BASS directly performs segmentation in the $14 \times 14$ feature map. This approach effectively smooths the boundaries and initially reduces the noise in nuclei boundaries. Furthermore, to minimize the impact of boundary uncertainty on segmentation, we apply a weighted loss to the low-resolution segmentation head. Specifically, pixels in the boundary region are assigned a lower weight, and other areas are set to a high weight.

According to previous studies (Wang et al., 2022), although low-resolution images can reduce the boundary noise, they lose some detailed information. To preserve the details, we parallel the original segmentation head and low-resolution prediction head to perform the segmentation task simultaneously, as shown in Fig.1(b). In this manner, the output mask head decreases the influence of the original feature noise and keeps more details.

## 2.4. Cross-RoI Contrastive Learning

In the subsection, we propose an elaborate denoising method named cross-RoI contrastive learning. It leverages labeled data to train a boundary feature extraction module, and then the module is applied to learn the embedding of unlabeled data, which can mitigate the impact of boundary noise caused by pseudo-labels and enhance the feature discrimination ability of foreground and background. In general, object boundaries typically correspond to hard-to-classify samples, and their embeddings are highly unstable. To avoid the impact of

features from difficult samples on the representation of easy-to-classify samples, we employ region-based contrastive learning and our proposed CRC is shown in Fig.1 (c).

First, the input image $x$ is fed into the network for feature extraction and alignment, then we randomly sample two aligned RoI features $f_i$ and $f_j$.

Based on the contour (plotted in black line) shown in Fig.1 (c), we split the feature maps into foreground region $\mathcal{F}$ and background region $\mathcal{B}$. By shrinking and expanding $d$ distances, we obtain the inner contour (plotted in red dotted line) and outer contour (plotted in blue dotted line). The regions between them and the true contour are represented as the inner boundary and outer boundary. These boundary regions correspond to challenging pixels for classification and can be expressed using the following formula:

$$
\begin{aligned}
\mathcal{R}_i &= \{p_i|\ p_i \in \mathcal{F}\ and\ \|p_i, c_i\|_2^2 \le d\}, \\
\mathcal{R}_o &= \{p_i|\ p_i \in \mathcal{B}\ and\ \|p_i, c_i\|_2^2 \le d\},
\end{aligned}
\tag{3}
$$

where $c_i$ represents the contour pixel closest to pixel $p_i$.

At the same time, we also sample other foreground and background pixels, which can be expressed as the following equations:

$$
\begin{aligned}
\mathcal{R}_f &= \{p_i|\ p_i \in \mathcal{F}\ and\ p_i \notin \mathcal{R}_i\}, \\
\mathcal{R}_b &= \{p_i|\ p_i \in \mathcal{B}\ and\ p_i \notin \mathcal{R}_o\},
\end{aligned}
\tag{4}
$$

where $\mathcal{R}_f$ represents the set of pixels obtained by excluding $\mathcal{R}_i$ from $\mathcal{F}$ and $\mathcal{R}_b$ represents the set of pixels obtained by excluding $\mathcal{R}_o$ from $\mathcal{B}$.

Next, we sample pixel features from the sets $\mathcal{R}_i$, $\mathcal{R}_o$, $\mathcal{R}_f$ and $\mathcal{R}_b$. The sampling ratio is set to $\alpha$. For feature $f_i$, the sampled features are denoted as $k_b^i$, $k_o^i$, $k_f^i$ and $k_i^i$ respectively. Similarly, for feature $f_j$, the sampled features are denoted as $k_b^j$, $k_o^j$, $k_f^j$ and $k_i^j$ respectively.

Then, we calculate the query features of background and foreground across RoIs through:

$$
q^b = M(k_b^i, k_o^i, k_b^j, k_o^j), \quad q^f = M(k_f^i, k_i^i, k_f^j, k_i^j),
\tag{5}
$$

where $M$ represents the averaged operation of vectors.

Finally, to narrow the same category feature distance and expand the feature distance between foreground and background. We calculate four pairs of contrastive losses as follows:

$$
L_{cl}^s = CL(q^b, k_b, k_f) + CL(q^b, k_o, k_i) + CL(q^f, k_f, k_b) + CL(q^f, k_i, k_o),
\tag{6}
$$

where $CL$ represents contrastive learning loss. $k_b$ represents the concatenation of $k_b^i$ and $k_b^j$. The calculation of $CL$ is described below:

$$
CL(q^+, k^+, k^-) = -log\frac{e^{cos(q^+,k^+)/\tau}}{e^{cos(q^+,k^+)/\tau} + \sum_{i=1}^{N} e^{cos(q^+,k_i^-)/\tau}},
\tag{7}
$$

where $q^+$ and $k^+$ represent a pair of positive instances, $k^-$ represents a negative instance, and $\tau$ is the temperature hyper-parameter.

Different from the previous methods, PC$_2$Seg (Zhong et al., 2021) extracts positive instance pair of contrastive learning from a single perspective. However, our proposed CRC performs contrastive learning cross-RoI, which enhances the feature generality.

Zhang Wang[1] Wang[2] Bian[3] Cai[1] Li[4] Zhang[3] Zhang[1,✉]

## 3. Experiments

### 3.1. Datasets

Our method is evaluated on the Cryosectioned Nuclei Segmentation (CryoNuSeg) dataset (Mahbod et al., 2021), the Digestive-System Pathological Segmentation (DigestPath) dataset (Da et al., 2022), and Multiple Organs Nuclei Segmentation (MoNuSeg) dataset (Kumar et al., 2017). CryoNuSeg contains 30 images from 10 organs, each with a size of $512 \times 512$. DigestPath contains 69 images of the digestive system, each with a size of approximately $1500 \times 1200$. MoNuSeg contains 30 images from 7 organs, each with a size of $1000 \times 1000$.

Table 1: Performance comparisons on CryoNuSeg, DigestPath and MoNuSeg Datasets. The best performance is highlighted in **bold**, and the second-best is underlined. † represents p-value of AJI < 0.001 and ‡ reprsents p-value of AJI < 0.05.

| Ratio | Methods | CryoNuSeg | | | DigestPath | | | MoNuSeg | | |
|---|---|---|---|---|---|---|---|---|---|---|
| | | Dice | AJI | PQ | Dice | AJI | PQ | Dice | AJI | PQ |
| 1/8 | Mask R-CNN† (He et al., 2017) | 50.28 | 26.43 | 27.17 | 52.58 | 29.12 | 30.87 | 70.03 | 48.76 | 45.29 |
| | MMT-PSM‡ (Zhou et al., 2020a) | 54.83 | 30.17 | 29.81 | 55.68 | 32.34 | 35.94 | 73.28 | 50.14 | 47.27 |
| | PointWSSIS† (Kim et al., 2023) | 58.66 | 35.41 | 33.61 | 59.90 | 40.06 | 44.10 | 76.12 | 50.80 | 49.34 |
| | ShapeProp‡ (Zhou et al., 2020b) | 57.42 | 35.53 | 33.68 | 58.18 | 39.94 | 43.49 | 75.33 | 49.89 | 50.02 |
| | NoisyBoundary† (Wang et al., 2022) | 55.14 | 29.57 | 30.96 | 58.34 | 36.75 | 37.94 | 75.27 | 48.14 | 49.37 |
| | PG-FANet ‡ (Jin et al., 2024) | 59.06 | 35.47 | 32.96 | 58.20 | 40.32 | 44.12 | 75.17 | 50.44 | 51.05 |
| | BASS† (Ours) | **59.26** | **36.32** | **35.09** | **61.00** | **41.33** | **45.07** | **77.43** | **51.80** | **53.05** |
| 1/4 | Mask R-CNN‡ (He et al., 2017) | 62.89 | 34.17 | 32.96 | 53.44 | 35.12 | 38.79 | 72.30 | 49.30 | 47.21 |
| | MMT-PSM‡ (Zhou et al., 2020a) | 67.24 | 37.60 | 34.67 | 58.23 | 37.64 | 41.93 | 73.14 | 51.08 | 49.17 |
| | PointWSSIS† (Kim et al., 2023) | **75.01** | 47.12 | 49.83 | **64.93** | 43.16 | 47.86 | 75.21 | 51.11 | 52.06 |
| | ShapeProp‡ (Zhou et al., 2020b) | 73.37 | 48.70 | 48.72 | 63.31 | 43.35 | 48.44 | 74.86 | 51.29 | 52.44 |
| | NoisyBoundary† (Wang et al., 2022) | 69.34 | 38.85 | 35.91 | 61.15 | 40.77 | 45.74 | 73.13 | 50.77 | 51.94 |
| | PG-FANet † (Jin et al., 2024) | 74.54 | 47.80 | 49.93 | 63.24 | 43.71 | 48.76 | 75.21 | 52.19 | 53.33 |
| | BASS† (Ours) | 74.79 | **48.96** | **50.36** | 63.41 | **44.72** | **49.14** | **76.34** | **53.39** | **55.85** |
| 1/2 | Mask R-CNN‡ (He et al., 2017) | 69.31 | 43.34 | 42.10 | 57.17 | 38.01 | 42.44 | 74.92 | 50.28 | 50.26 |
| | MMT-PSM‡ (Zhou et al., 2020a) | 72.85 | 45.06 | 44.47 | 59.11 | 39.97 | 45.58 | 75.12 | 51.05 | 51.17 |
| | PointWSSIS† (Kim et al., 2023) | 74.67 | 49.91 | 49.29 | 63.87 | 45.45 | 51.64 | 75.89 | 52.14 | 52.30 |
| | ShapeProp‡ (Zhou et al., 2020b) | 74.40 | 48.24 | 47.55 | 64.15 | 45.02 | 52.93 | 76.01 | 51.88 | 52.94 |
| | NoisyBoundary† (Wang et al., 2022) | 73.71 | 46.58 | 46.13 | 61.35 | 44.41 | 50.65 | 77.10 | 53.99 | 55.20 |
| | PG-FANet ‡ (Jin et al., 2024) | 72.17 | 49.86 | 49.37 | 64.49 | 45.14 | 51.15 | **78.77** | **54.91** | 56.04 |
| | BASS† (Ours) | **76.76** | **51.09** | **49.66** | **65.72** | **46.14** | **53.96** | 77.80 | 54.82 | **56.59** |

### 3.2. Implementation Details and Evaluation Metrics

Following the previous method (Graham et al., 2019), we crop all the images to patches of $256 \times 256$ pixels with an overlap of 128 pixels for data preprocessing. All experiments are carried out with an RTX 3090 GPU. SGD is used as the optimizer. The learning rate, momentum, and weight decay are set to 0.02, 0.9, and 0.001, respectively. Besides, we evaluate the segmentation performance in terms of Dice (Vu et al., 2019), aggregated Jaccard index (AJI) (Kumar et al., 2017), and panoptic quality (PQ) (Kirillov et al., 2019).

### 3.3. Comparison with the State of the Art Methods

We compare our proposed BASS against several state-of-the-art methods, including MMT-PSM (Zhou et al., 2020a), PointWSSIS (Kim et al., 2023), ShapeProp (Zhou et al., 2020b), NoisyBoundary (Wang et al., 2022) and PG-FANet (Jin et al., 2024). Besides, to validate

the improvement of our semi-supervised model, we also compare our method with supervised Mask R-CNN. We trained the models using 1/8, 1/4, and 1/2 of the labeled data on ResNet-50. A more detailed data split is presented in the appendix.

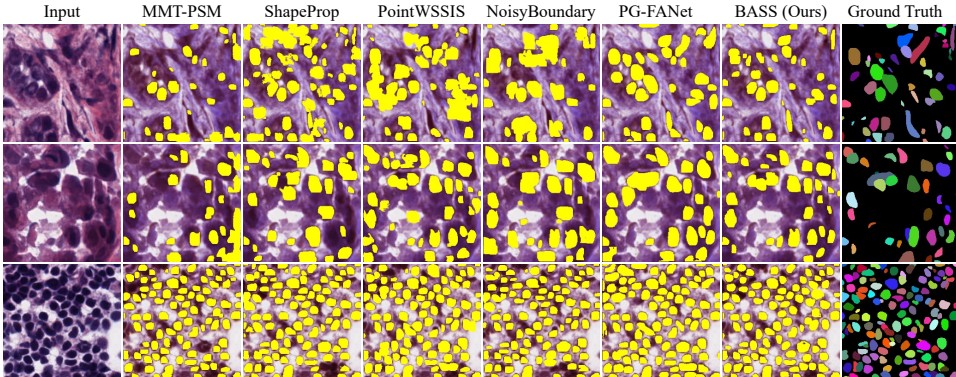

Figure 2: The semi-supervised instance segmentation visualization comparisons.

Quantitative comparison results on three datasets are displayed in Table 1, which shows that our method achieves the optimal performance at all three annotation ratios. Even with only 1/8 of the annotations, our BASS exceeds the suboptimal method approximately 1% in PQ. Fig.2 displays the visual comparison results. We can see that MMT-PSM and NoisyBoundary mistakenly identify nuclei as tissue due to the lack of semantic discrimination between nuclei and tissues. Although ShapeProp and PointWSSIS employ weak labels to enhance the location ability of nuclei, they still have nuclear shape errors.

Table 2: The segmentation head ablation experiments on CryoNuSeg and DigestPath.

| NMH | LRD | CRC | CryoNuSeg | | | DigestPath | | |
|:---:|:---:|:---:|:---:|:---:|:---:|:---:|:---:|:---:|
| | | | **Dice** | **AJI** | **PQ** | **Dice** | **AJI** | **PQ** |
| ✓ | | | 65.49 | 40.23 | 36.94 | 60.80 | 44.73 | 48.54 |
| | ✓ | | 64.37 | 39.24 | 39.91 | 61.22 | 43.09 | 47.80 |
| ✓ | ✓ | | 72.17 | 46.44 | 45.72 | 63.19 | 45.86 | 51.22 |
| ✓ | ✓ | ✓ | **76.76** | **51.09** | **49.66** | **65.72** | **46.14** | **53.96** |

### 3.4. Ablation Studies

**Ablation Studies for Segmentation Head.** In the student network, we employ three prediction heads, namely, the naive mask head (NMH), low-resolution denoising mask head (LRD), and cross-RoI contrastive learning mask head (CRC), to jointly supervise the segmentation predictions. To evaluate the effectiveness of these heads, we conducted a series of ablation experiments to assess the impact of different designs. Specifically, we compared four designs: NMH, LRD, NMH+LRD, and NMH+LRD+CRC. The experimental results are listed in Table 2. From the table, we can observe the NMH+LRD+CRC outperforms the other methods, indicating that incorporating multiple segmentation constraints is effective.
**Ablation Studies for Box and Pixel Thresholds** We conduct threshold filtering experiments on the MoNuSeg dataset with a 1/2 annotation ratio, and The experiment results are shown in Tables 3 and 4. Table 3 uses nuclear count statistics to determine the opti-

Zhang Wang[1] Wang[2] Bian[3] Cai[1] Li[4] Zhang[3] Zhang[1,✉]

mal box threshold. We can find that the model performs best when the threshold is set to 0.38. When changing the value, the model's performance deteriorated, indicating the effectiveness of using nucleus count for box threshold selection. In addition, Table 4 shows the experiment results of pixel threshold. The table shows that when choosing 0.5 as the threshold, the model achieved optimal performance in terms of Dice and PQ scores.

Table 3: The box threshold setting experiments.

| Box Thr | Dice | AJI | PQ |
|---|---|---|---|
| 0.3 | 76.37 | 52.20 | 54.07 |
| **0.38(opt)** | **77.80** | **54.82** | **56.59** |
| 0.5 | 75.68 | 50.83 | 53.26 |
| 0.7 | 72.28 | 48.01 | 51.58 |

Table 4: The mask threshold setting experiments.

| Pixel Thr | Dice | AJI | PQ |
|---|---|---|---|
| 0.3 | 76.12 | 54.01 | 55.23 |
| 0.4 | 77.04 | **54.96** | 55.37 |
| **0.5(opt)** | **77.80** | 54.82 | **56.59** |
| 0.6 | 76.97 | 54.04 | 56.21 |

**Ablation Studies for Sampling Ratio $\alpha$.** We conduct sampling ratio experiments and set four sampling ratios of 0.1, 0.3, 0.5, and 0.7 in the CRC. The experimental results are shown in Fig. 3. The table shows that as the sampling ratio increases, the performance gradually improves, indicating that the sampling ratio indeed influences the performance. When the sampling ratio is large, the model obtains more sampled pixels, resulting in better contrastive learning performance. However, as the sampling ratio increases, the computational cost of the model also increases. Therefore, we select 0.7 as the final sampling ratio, which achieves the best balance between model performance and computational cost.

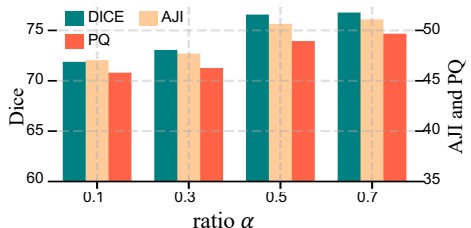

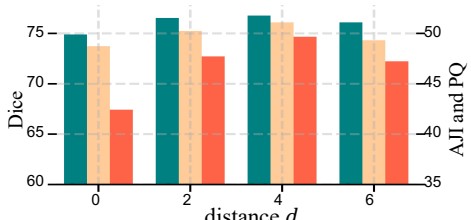

Figure 3: The sampling ratio ablation experiments on CryoNuSeg dataset.

Figure 4: The distance comparison experiments on CryoNuSeg dataset.

**Ablation Studies for Distance $d$.** We investigate the effect of distance $d$, which represents the distance from the inner (outer) contour to the accurate nuclei contour. Expressly, we set $d$ to 0, 2, 4, and 6. It is worth noting when $d = 0$, we do not sample between the actual and inner (outer) contour. From the Fig. 4, we can see when $d = 4$, the model performs best. However, the performance drops as $d$ decreases. This is because when reducing the sampling range, the boundary information obtained by the model also decreases. On the contrary, when $d$ increases to 6, the sampling area becomes more extensive, leading to a mixture of boundary and non-boundary features, ultimately decreasing performance.

## 4. Conclusions

This paper proposes a boundary-aware contrastive learning model based on the teacher-student framework for semi-supervised nuclei segmentation. The model utilizes a low-resolution feature supervision head and a cross-RoI contrastive learning module to achieve the nuclei boundary denoising. However, the model trains the teacher and student networks in separate stages, which hinders the student network from effectively utilizing the features

extracted by the teacher network. Therefore, in the future, we will adopt an end-to-end training approach for both the teacher and student networks to enhance the information interaction between the teacher and student networks.

## 5. Acknowledgements

This work was supported in part by the National Natural Science Foundation of China under 62031023 & 62331011; in part by the Shenzhen Science and Technology Project under GXWD20220818170353009, and in part by the Fundamental Research Funds for the Central Universities under No.HIT.OCEF.2023050.

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

## 6. Appendix

In the main body, we used 1/8, 1/4 and 1/2 labeled data to conduct experiments on CryoNuSeg (Mahbod et al., 2021), DigestPath (Da et al., 2022) and MoNuSeg (Kumar et al., 2017) datasets respectively.

In this section, we provide the data split details as shown in Table 5. First, these three datasets are divided into the training set, validation set and testing set according to the proportion of 6:2:2. Then, we re-divide the training set into labeled and unlabeled data sets according to 1/8, 1/4 and 1/2. In the whole training process, we keep the validation and testing sets unchanged.

Table 5: The data split on CryoNuSeg, DigestPath and MoNuSeg datasets.

| Dataset | Ratio | Training | | Validation | Testing |
|---|---|---|---|---|---|
| | | Labeled | Unlabeled | | |
| **CryoNuSeg** | **1/8** | 20 | 142 | 54 | 54 |
| | **1/4** | 40 | 122 | 54 | 54 |
| | **1/2** | 81 | 81 | 54 | 54 |
| **DigestPath** | **1/8** | 631 | 2653 | 835 | 994 |
| | **1/4** | 930 | 2354 | 835 | 994 |
| | **1/2** | 1740 | 1554 | 835 | 994 |
| **MoNuSeg** | **1/8** | 98 | 686 | 392 | 294 |
| | **1/4** | 196 | 588 | 392 | 294 |
| | **1/2** | 392 | 392 | 392 | 294 |

