# OpenReview forum: "Boundary-aware Contrastive Learning for Semi-supervised Nuclei Instance Segmentation"
_MIDL.io/2024/Conference — MIDL 2024 Oral_

### Official Review · Reviewer_yGzw · 2024-02-15

**Confidence:** 5
**Preliminary Rating:** 4
**Final Rating:** 4

**Summary:**

The study introduces a novel contrastive learning model, designed with an awareness of boundaries, for the task of semi-supervised nuclei segmentation, employing a teacher-student framework. Within the framework, the student network incorporates a low-resolution denoising module alongside a cross ROI contrastive learning module. These modules are specifically aimed at reducing the contour noise of nuclei from both macro and micro perspectives. Comprehensive comparative analyses conducted on two publicly available datasets have demonstrated the enhanced performance of this approach over existing methods in semi-supervised instance segmentation.

**Strengths:**

+ Design of a low-resolution denoising (LRD) segmentation head to enhance boundary smoothness.
+ Implementation of a low-weight loss within the segmentation head for optimizing nuclei boundary regions, reducing the effects of uncertain boundary predictions during training.
+ Creation of a cross-RoI contrastive learning (CRC) module to further diminish boundary noise by finely distinguishing between the internal, external, and boundary regions of nuclei, thereby improving the discriminative power of nuclei boundary features.
+ Validation of the proposed method's effectiveness through comparative experiments and ablation studies on two public datasets.

**Weaknesses:**

The review highlights a significant shortfall in the evaluation's thoroughness, noting its failure to compare the segmentation approach with current state-of-the-art (SOTA) models, despite using public datasets. This gap makes it difficult to accurately assess how the proposed method stacks up against the leading techniques in the field.

Moreover, the analysis lacks a detailed statistical examination. The absence of statistical significance tests, like p-values, raises concerns about the reliability of the differences observed between the proposed method and others. Such analysis is essential in medical image research to ensure findings are not due to chance.

The critique also questions the adequacy of the training sample size for a semi-supervised contrastive learning approach, suggesting that a directly supervised learning method might have been more appropriate. Furthermore, the performance metrics presented, particularly the Dice scores, are considered low. This observation underscores the need for a comparison with fully-supervised methods, such as Mask-RCNN, to provide a clearer understanding of the proposed method's efficacy.

**Detailed Comments:**

In the preceding sections, I have discussed both the strengths and weaknesses of the work. Beyond these points, there is additional scope for enhancing the clarity and precision of the language used in the document. Improvements in linguistic expression could further refine the presentation and comprehension of the research, making it more accessible and impactful to its intended audience.

**Justification Of Final Rating:**

Thanks very much for the author's response. The paper's quality has significantly improved. My major concerns have been resolved, especially the statistical analysis. Therefore, I continue to support my positive evaluation.

**Justification Of The Preliminary Rating:**

The paper demonstrates notable strengths, such as the introduction of innovative methods and their contributions to the field, underscoring its potential significance and utility in its domain, as previously discussed. However, it also has its drawbacks, including concerns with the evaluation methodology, statistical rigor, and the rationale behind selecting certain approaches over others, as detailed earlier. These shortcomings bring into question the thoroughness and reliability of the research outcomes. Considering these strengths and weaknesses collectively, the decision lean to positive

**Questions To Address In The Rebuttal:**

In outlining the key issues in the weaknesses section, I've identified areas that, if improved upon, could substantially elevate both the quality and influence of the paper. Addressing these points would be greatly advantageous for the authors.

---

> ### Author Response · Authors · 2024-03-17
> **Reply to Reviewer yGzw**
>
> Thank you for your recognition of our article and your valuable comments. We will reply to your suggestions individually in the hope that these answers can solve your confusion.
>
> Weaknesses
>
> (1) According to your suggestion, we added the latest semi-supervised nuclear comparison method, PG-FANet, and the experiment results are shown in Table 1 of the main text. The experiment results show that the performance of our method is superior to that of state-of-the-art, which confirms the effectiveness of our method.
>
> [1] Jin Q, Cui H, Sun C, et al. Inter-and intra-uncertainty based feature aggregation model for semi-supervised histopathology image segmentation[J]. Expert Systems with Applications, 2024, 238: 122093.
>
> (2) Thanks for your valuable advice, we did neglect statistical testing, which is very important for the nuclear segmentation task. Based on your comments, we evaluated each method for statistical significance. In Table 1 of the main text, we have added the identification of the significance test. Thank you again for your professional advice.
>
> (3) Thanks for your constructive comments; our model is feasible with a small sample size. This is because our contrastive learning is performed on pixel features. At each iteration, the model will randomly sample a fixed proportion (α=0.7) from a mini batch for contrastive learning. Due to randomness, the samples sampled during each iteration are not the same, so a small number of training samples is enough for model training.
> Your comments are valuable, full supervision is a suitable application, and our model can work. In the Table 6 of supplement materials we give a fully supervised experiment, and the experiments show that contrastive learning is effective for fully supervised learning. In the future, we will enhance our contrastive learning program to achieve better performance in both fully supervised and semi-supervised tasks.
>
> (4) The low Dice index is caused by two reasons: (1) First, we adopted stricter data partitioning when training the model. We split the training, validation, and testing sets on a 6/2/2, while many articles report results on 8/1/1 or 7/1.5/1.5. (2) Second, it is challenging to conduct a semi-supervised model on CryoNuSeg and DigestPath datasets, which not only have a small size nuclei morphology, but also have a very significant difference in nuclear morphology (CryoNuSeg id a pan-cancer dataset) between the training set and the test set.
> In addition, according to your suggestion, we conducted supervised experiments on Mask R-CNN (shown in Table 1 of main text) and the experimental results also showed that the Dice value on CryoNuSeg data was low, but on the whole, our semi-supervised method was still greatly improved compared with the supervised model.
>
> Questions To Address In The Rebuttal
>
> Thank you for your constructive comments and professional guidance on this article. According to your comments in weakness, we have explained these problems one by one in our reply. At the same time, we also added some comparison experiments and revised the methods in the main body and supplementary materials to make the paper read more smoothly. We hope these replies will satisfy you. If you have any other questions, you are welcome to point out, we will make further positive reply.
>
> Justification Of The Preliminary Rating
>
> Thank you again for your recognition of the method and pointing out the shortcomings of our work. Your suggestions have positive guiding significance for our future work, and we will continue to improve this work according to your suggestions.

---

> > ### Comment · Reviewer_yGzw · 2024-03-21
> >
> > Thanks for the response from the author. The quality of the paper has been substantially improved. My concerns have been addressed. So I still render the positive decision.

---

### Official Review · Reviewer_mLet · 2024-02-21

**Confidence:** 4
**Preliminary Rating:** 2
**Recommendation:** Poster

**Summary:**

This paper proposes a boundary-aware contrastive learning model for semi-supervised nuclei segmentation based on the teacher-student framework. In the student network, a low-resolution denoising module and a cross-RoI contrastive learning module are proposed to ease the contour noises of nuclei from coarse and fine aspects.

**Strengths:**

The paper is easy to follow and well-organized.
The problem seems importance as labeling the nuclei is very laborious
The method is validated on two datasets and shows promising performance compared to some baselines.

**Weaknesses:**

I have some concerns regarding the paper.
1.	The cross-RoI contrastive learning needs a pretrained boundary feature extraction module which is said to train on labeled data. So, does the method need a 2-stage training? If yes, this will make this method complicated, if I change the ratio of labeled data, I need to retrain a boundary feature extraction module. This will also pose another issue: the method will heavily depend on the trained feature extraction module, and the performance of this module will also depend on the quantity and quality of your selected labeled data. So, how to ensure the quality of this feature extraction module? Especially under extremely limited labeled conditions.
2.	Will the denoising only preserve easy cases? Like you have filtered out most of the pixels via box- and pixel-level thresholds. Yet, the filtered pixels can provide more information for training as they may be the hard cases. Will this lead to bias?
3.	More semi-supervised baselines should be adapted to compare. Many methods focused on pseudo-labeling.
4.	What are the weak and strong augmentation used in the paper?
5.     The computation cost should be listed and compared to previous methods.

**Detailed Comments:**

See above

**Justification Of The Preliminary Rating:**

1.	The cross-RoI contrastive learning is very complicated and may be unstable due to several factors I have mentioned.
2.	Some details should be clarified as mentioned.
3.	Computation cost is also an important issue.

**Questions To Address In The Rebuttal:**

See above

**Special Issue:**

No

---

> ### Author Response · Authors · 2024-03-17
> **Reply to Reviewer mLet**
>
> We sincerely appreciate your suggestions regarding the methods and experiments presented in this paper. We have thoroughly analyzed your questions and provided detailed responses to address them.
>
> Weaknesses
>
> (1) We sincerely apologize for any confusion caused. We would like to clarify that the two-stage training is not necessary for calculating cross-RoI CL. Our CL is trained in an end-to-end manner. However, during the loss calculation, we only utilize labeled data to compute the contrastive loss, even though the unlabeled (pseudo-labeled) data also generate RoIs.
> In our cross-RoI CL, the boundary feature extraction is carried out on the generated RoIs, which is the outputs of the Mask R-CNN detection step. As described in 'unified reply', the Mask R-CNN is trained in an end-to-end manner and the detection results can be revised in the second step. Therefore, a different labeled ratio don't need train the network from scratch and a small number of samples will not directly impact the feature extraction. Because the segmentation network will continue to correct the detection results.
>
> Also, about ``whether the method you mentioned will depend on the quantity ..."., which can be answered in two ways: (1) In the original Mask RCNN, to enhance the feature extraction capability and consider the size change of the object, the author adopts FPN in the feature extraction network. (2) Since the model be trained end-to-end, and total loss is weighting multiple loss, so the network can continue to optimize objects with inaccurate location in RPN. Therefore, the process will adjust those ROIs to make the prediction closer to the actual object position.
>
> To provide you with a better understanding of our training method, we would like to further elaborate on the details.  Mask-RCNN is a two-stage detection framework. Under mmdetection, detectron or other frameworks, Mask-RCNN can be trained end-to-end. Firstly, an RPN network is used to locate the objects, in which a large number of RoI regions are generated. Secondly, a segmentation network makes segmentation predictions for each extracted RoI feature. Our cross-RoI contrastive learning module is improved based on the existing structure, which does not require an additional training process but only uses the features extracted from RoI. We sincerely hope that these answers have addressed your concerns and cleared up any confusion.
>
> (2) Thank you for your question. In the original article, we lacked a description of this part. Now, we add box and pixel thresholds in Section 2.2 of main text, we hope it can help readers understand our approach. For question "the filtered pixels can provide more information ...", which does exist. But, we can only minimize the influence of threshold selection on the generation of pseudo-labels, so we use the iteration optimization strategy. To facilitate your understanding, we have added the threshold setting in Section 2.2. Besides, to verify the effectiveness of our method, we supplement ablation experiment in Table 3-4. The experiment proves that the threshold optimization is better than the method of setting hyperparameter in box threshold.
>
> (3) This is a good suggestion. We have added the latest state-of-the-art method PG-FANet in the Table 1 of main text.
>
> (4) Following UniMatch, CCVC, and other methods, in SSL, data augmentation and consistency constraints always are used to improve the segmentation performance. Generally, weak data augmentation includes image shape changes, while strong data augmentation includes image color jitter. In this article, we use image resize and rotation for weak data augmentation; strong data augmentations are Gaussian noise and image blur.
>
> (5) Thank you for your professional advice. We have added the parameters quantity comparison experiments in the Table 9 of supplement materials.
>
> Justification of XX
>
> (1) Contrastive loss instability does affect training. For this, we visualize the contrastive loss in Fig. 5 of supplement materials, which shows the loss has apparent downward and convergence trend. That's because we only sample part pixels to calculate the CL and take the average operation on the query vector, which can ensure the feature better represent the general features of foreground and background. Therefore, the loss curve is smooth.
>
> (2) According to your suggestions, we answered these detailed questions individually as shown above.
>
> (3) We compared the computation costs of baseline model in the Table 9 of supplementary material. If any other problems need to be addressed please point out we will further explain or improve.
>
> We hope our responses can address your questions and aided in your understanding of the content presented in this article. If you still have any concerns or issues regarding the model design and experimental procedures, please bring them to our attention. Your feedback is valuable to us, and we are committed to addressing any remaining challenges or inquiries.

---

### Official Review · Reviewer_DPah · 2024-02-23

**Confidence:** 5
**Preliminary Rating:** 4
**Recommendation:** Poster
**Final Rating:** 5

**Summary:**

The authors propose a way to improve the segmentation of nuclei inside glomeruli using semi-supervised learning. The proposed method uses techniques based on pseudo-labeling and denoising of nuclei edges. They comparatively evaluated the method against other ones, achieving better performances on instance segmentation.

**Strengths:**

The paper is well-written and very well-organized. The proposed method is well-motivated in all aspects and ablation studies ground specific performance issues of parts of the method. A benchmark was made against other methods.

**Weaknesses:**

The authors relied on a notably limited dataset, with CryoNuSeg comprising only 30 images and DigestPath containing 69 images. Furthermore, they omitted any statistical significance study through cross-validation, which is crucial for robust evaluation.

Additionally, the conclusion appears to be merely a summary of the paper's content rather than a conclusive analysis or discussion of the findings.

**Detailed Comments:**

While the paper is indeed well-written, there are two notable areas that, in my view, warrant improvement: the method's generalization capability, particularly given the suboptimal DSC performance, and the conclusion section. I suggest rephrasing the conclusion to encompass discussions and ideas regarding these points for future research directions.

**Justification Of Final Rating:**

I commend the authors for carefully addressing all the issues raised in my review. With the addition of another dataset and a more thorough experimental analysis, I am revising my final rating to a strong acceptance.

**Justification Of The Preliminary Rating:**

The paper makes significant contributions to nuclei segmentation through the application of semi-supervised learning. It is well-structured and conducts pertinent experiments to assess the effectiveness of the proposed method.

**Questions To Address In The Rebuttal:**

How can you enhance the generalizability of your experiments to more accurately assess this aspect?

**Special Issue:**

No

---

> ### Author Response · Authors · 2024-03-17
> **Reply to Reviewer DPah**
>
> Weaknesses
>
> (1) Thank you for your reply and pointing out the shortcomings of our experiment, which is extremely important for presenting our experimental results.
> About the amount of data, though the amount of DigestPath data is small, the size of each labeled region is large. In data preprocessing, the overlapping strategy is adopted to split images, and about 5000 patches of 256*256 are obtained on the DigestPath dataset. We consider that this data is enough for model training and validation. At the same time, according to your suggestion, we added a new MoNuSeg dataset to perform experiments, and the results are shown in Table 1 of the main text, which further confirmed the effectiveness of our proposed method. In addition, we supplemented the p-value statistical significance experiment in the Table 1 of mian text, and the experimental results showed that our method met the statistical significance hypothesis. Furthermore, we performed a 4-fold cross-validation experiment and calculated the standard deviation of cross-validation. Due to the limited length of the text, we present the experimental results in the Table 7 of supplementary materials. Thank you again for your suggestions. You are welcome to point out any inadequacies and we will actively revise them.
>
> (2) Thank you for your valuable suggestions. We have read the conclusion carefully, and there are indeed problems you mentioned. According to your suggestions, we have rewritten the conclusion section and expounded the possible improvement of the article in the future. If there is still any need for improvement, please point it out, and we will actively revise it according to your comments.
>
> Detailed Comments
>
> Thank you for your constructive comments. Based on your comments, we have carried out two generalized verification aspects.
> First, we conducted a generalization experiment of the model. Specifically, we performed experiments and comparisons on the new data MoNuSeg, and the results are shown Table 1 of main text. From this result, we can see that our method still performs best on the latest data set, confirming that our model has good generalization.
> Secondly, we conducted generalization experiments on the parameters of the model. Specifically, we mutually verified the model performance on CryoNuSeg and DigestPath. Due to the limited length of the text, we present the experimental results in the Table 8 of supplementary materials. The experimentresults show that when our model is test on other dataset, the performance decreases. This phenomenon is because our method does not consider distribution differences among datasets, resulting in a poor cross-data set validation effect.
> Finally, according to your suggestion, we modified the conclusion and marked it with blue font in the body. Specifically, we briefly summarize our method and illustrate the shortcomings of our method. This is also the improved direction of our model in the future. Thank you again for your valuable suggestions in our work.
>
> Questions To Address In The Rebuttal
>
> (1) For the generalization problem of the model, we performed generalization experiments on the model and parameters, and the experimental results showed that our model had good generalization performance on the data set. Still, it did have shortcomings in generalization across the data set. We will improve this problem in the future. For example, strong data augmentations help the model train on richer images and feature consistency constraints are used to enhance the generalization performance of features extracted by the model.
>
> Justification Of The Preliminary Rating
>
> (1) Thank you for your comments on the writing and experiments of this paper. We have made careful revisions based on your comments. If there are still other problems in the paper, please point them out, and we will continue to improve them. Thanks again for your comments on this article.

---

### Official Review · Reviewer_8Bvg · 2024-02-29

**Confidence:** 4
**Preliminary Rating:** 3
**Final Rating:** 4

**Summary:**

This paper proposes a method for nuclei segmentation in a semi-supervised setting, using a teacher-student model, with extra regularization involved.

The results seem good, on two different datasets, but overall the paper is difficult to follow and to fully understand the complete method.

**Strengths:**

- The results are seemingly good, outperforming the chosen baselines.
- Many ablation and sensitivity studies are performed, with parts of the method and different percentages of labeled data.
- Several methods are compared to, on two different datasets.

**Weaknesses:**

- The method is quite complex and not all parts are necessarily motivated when introduced
- The paper is raher difficult to follow (out of the 5 papers I reviewed this is the one that gave me the most troubles to make sense of), partly because is written as code and not as actual mathematical notation
- ironically, the code is not shared. For such a complex method, I do not think that it could be reproduced without access to the code.

**Detailed Comments:**

- "In the paper, $\omega_1,\omega_2,\omega_3$ are set to 1" Then simply do not add it in the notation
- Simply rewrite $P_{fore} \setminus P_{inn}$ instead of $P_{fore-inn}$
- Using the features of LaTeX for math notation, with alternative fonts and alphabet, could greatly help the notation.
- DSC and AJI are virtually the same metric

> To mitigate the impact of boundary noise, BASS employes a coarse denoisining strategy. Firstly, we apply box and pixel thesholds to filter the predicted results.

This is repeated several times before being actually explained. The manuscript could therefore probably be reworked to avoid such repetition.

**Justification Of Final Rating:**

The authors have significantly improved the manuscript and clarity, and also shared the public code in the meantime. Congratulation for the good and hard work performed in such a short time.

This addresses most of my initial concerns about the method. DSC and AJI are still pretty much the same metric but that could easily be addressed in a journal extension.

I recommend acceptance of this paper, as a poster.

**Justification Of The Preliminary Rating:**

While the performances _seem_ to be there, the fact that the method is so complex (and its explanation could be improved) make me wary to commit ot a higher rating. I truly feel that I could be missing something very important, and that ultimately the method could be much simpler and elegant.

So this will need to be clarified and sorted out in the rebuttal. I do not believe that pure performances are enough for acceptance, if the method involved is so complex that no-one can re-implement and re-use it.

**Questions To Address In The Rebuttal:**

- Not quite a question, but I think that as the rebuttal/discussion period lasts three weeks, there is plenty of time to rework and re-explain the core of the method.
- What are the box and pixel threshold precisely? Do they involve hyperparemeters to set manually?

---

> ### Author Response · Authors · 2024-03-17
> **Reply to Reviewer 8Bvg**
>
> Weakness
>
> (1) Thank you for your questions regarding the method described in the article. We have had a thorough discussion after carefully considering your comments. The problem you have raised may be caused by our inadequate description of the baseline method Mask R-CNN, for which we have added a description of the teacher model baseline in the revised vision in the Section 2.1 Line 90-92.
>
> The modules proposed in this paper build upon the existing structure of Mask R-CNN. The method itself is designed to be straightforward and approachable. We have taken care to ensure that the proposed modifications and additions are implemented in a manner that maintains clarity and simplicity.
>
> In the teacher model, we use Mask R-CNN as the baseline, a two-stage detection and segmentation method, which can be trained end-to-end. The first stage of the model is responsible for detecting and outputting some candidate box positions and corresponding confidence scores. The second stage of the model performs segmentation using the detection results, and our LRD and CRC approaches are designed based on the existing framework.
>
> In the student model, we retrain a Mask R-CNN model using data with annotated labels and pseudo-labels. We introduce low-resolution denoising and contrastive learning modules into the segmentation head to achieve boundary denoising.
> In the paper, the generation of pseudo-labels does not require training, so the model only needs to train a teacher network and a student network. As for training methods, many wrapped deep learning frameworks can directly serve the segmentation or detection training, such as Mask R-CNN, FCOS, and so on. It is easy to implement the method by directly invoking these libraries (e.g. mmdetection and detectron2).
>
> In addition, based on your comments, we reviewed the method described in this article again. Just as you said, there are some redundant expressions in the opening sentence of each paragraph. We have revised the statement in blue font. Please let us know if you have any questions, and we will actively respond to them. The revised version has been uploaded to openreview.
>
> (2) Thank you for your patient guidance and advice. Based on your suggestions, we have carefully revised the formula and made it as clear as possible. Specifically, we use more concise and distinguishable symbols to represent the four areas of CL, hoping this will reduce the difficulty of reading. In addition, you can also see the modification result in the main text Eq.3 and Eq.4 of Section 2.4.
>
> (3) Due to our oversight, we did not open source the code in the initial release. Now, we have uploaded the code to GitHub: https://github.com/zhangye-zoe/BASS. You can refer to the instructions in the README.md to run our work. For the introduction of each section, we also added specific explanations in the corresponding implementation parts of the code.
>
> Detailed Comments
>
> (1) Thank you for pointing out the problem in the article in detail. We have simplified the expression of the weight of the loss function, eliminating the w1, w2, and w3 weights.
>
> (2) According to your suggestion, we have improved this part of the formula. You can see the modified result in the annotations in the main text. Thank you for pointing out the problems in the article's expression.
>
> (3) Your suggestions have been beneficial to the revision of our article, so in the body, we have simplified some formula symbols, which will be easier for other readers to read.
>
> (4) As two important indexes for evaluating nuclear instance segmentation, Dice and AJI are similar to some extent. However, their emphasis is different. Dice pays more attention to pixel-level segmentation performance, while AJI tends to evaluate the quality of connected domain segmentation. Following previous works HoverNet and TripleU-Net, we also adopted Dice and AJI as evaluation criteria.
>
> (5) There are indeed many redundant expressions in the previous version, and we have corrected these repeated expressions. These changes have been highlighted in blue for your convenience.
>
> Questions To Address
>
> (1) Thank you for listing the shortcomings of this article. We have carefully read your suggestions and made detailed revisions, such as method description and formula expression. In addition, we added many comparison experiments and ablation experiments.
>
> (2) For methodological details of box and pixel threshold, because of limited page of reply, we added them Section2.2 of body. Besides, we validate the effectiveness of this threshold filtering in Section 3.4 ablation experiments.
>
> (3) Thank you for your feedback on our methods and experimental details. We have provided responses to address these issues, but there may still be some areas that need to be clarified in our replies. If there are any points that you still do not understand, please let us know, and we will be more than happy to provide further clarification.

---

### Author Response · Authors · 2024-03-17
**An Unified Reply to All Reviewers**

Thank you for your kind and valuable suggestions. We greatly appreciate your feedback, as it will contribute to the improvement of our article. In the part, I would like to give a unified reply to the questions that have been raised many times in the review. We hope that this unified response can help reviewers o have a general understanding of our modifications.

Firstly, regarding the code, previously, due to our negligence, we did not upload it. Now, we have uploaded the code to GitHub: https://github.com/zhangye-zoe/BASS. We believe that accessing the code will provide you with a better understanding of the methods discussed in the article. Meantime, based on comments of reviewers, we also uploaded the revised text to the following link: https://openreview.net/forum?id=rVx9DiR5Ha&referrer=%5BAuthor%20Console%5D(%2Fgroup%3Fid%3DMIDL.io%2F2024%2FConference%2FAuthors%23your-submissions)where we highlighted the revised text in blue.

Secondly, about the two questions: (1) box(pixel) threshold choose method (if it is chosen like hyperparameter) and (2) training method (when we conduct contrastive learning whether we need a single training or not). To answer these questions, we need to introduce the baseline Mask R-CNN.

In this paper, we use Mask R-CNN as the baseline model, and the network can be end-to-end training to achieve detection and segmentation. In the process of object detection, Mask-RCNN outputs some candidate boxes (RoIs) and corresponding confidence scores (box scores). In general, the model uses 0.5 as a threshold to filter RoIs with high confidence. However, given the characteristics of less data annotation and a large number of nuclei in semi-supervised tasks, the fixed threshold may not suitable for all scenarios of nuclear segmentation. Therefore, we used optimization strategies to choose the threshold values $v_b$ and $v_p$.  For your understanding, we added the method description of the box and pixel threshold in the Section 2.2 of main text. This process is simple but effective and the code is uploaded to the following file to facilitate your understanding: https://github.com/zhangye-zoe/BASS/blob/main/scripts/kumar/filter_pl.py.

For the cross-RoI contrastive learning, based on the extracted RoI features in the detection stage, we directly conduct contrastive learning on these RoI features. Thus, they don’t need a single training stage. If you have any further questions or require additional clarification, please let us know, and we will be glad to reply you.

Next, we genuinely appreciate the valuable feedback provided by the reviewers regarding the equation symbols and the conclusions of our article. We have carefully considered their suggestions and made the necessary corrections to ensure the expressions are explicit and easily comprehended. The revised version can be found at the following link: https://openreview.net/forum?id=rVx9DiR5Ha&referrer=%5BAuthor%20Console%5D(%2Fgroup%3Fid%3DMIDL.io%2F2024%2FConference%2FAuthors%23your-submissions). If there are any further areas you find unreasonable or require improvement, please kindly point them out, and we will continue to enhance the clarity and readability of the paper.

Finally, we sincerely thank all the reviewers for their insightful suggestions. We have considered their comments carefully and made significant additions to our experimental design. These additions include experiments conducted on the new dataset MoNuSeg, a new semi-supervised segmentation comparison method, cross-validation experiments, and a comparison of model parameters. To ensure that the results of these comparison experiments are accessible to everyone, we presented all the additional experiments in the supplement materials, including Table 6, 7, 8, 9 and Figure 5. Furthermore, we have addressed each reviewer's questions and concerns with thorough responses. If you have any specific questions regarding the methodology, we will happily respond comprehensively.

Once again, we extend our appreciation to reviewers for detailed suggestions, which have significantly enhanced our article's quality. We value their contributions and remain committed to delivering a comprehensive and robust research paper.

---

### Comment · Area_Chair_q5fX · 2024-03-17
**Please read and respond to author comments**

Dear reviewers. The authors have posted responses to your reviews. Please take the time to read and respond before March 27.

---

### Meta-Review · Area_Chair_q5fX · 2024-04-03

**Recommendation:** Accept (Poster)
**Confidence:** 5

**Metareview:**

This paper proposes a novel boundary-aware contrastive learning model semi-supervised nuclei segmentation. The reviewers found the paper to be well motivated with good results demonstrated by comparative experiments and ablation studies. There were some initial concerns around lack of statistical results pointing to the significance of the results, lack of SOTA methods in methods compared against, issues with clarity, code not being shared which all appear to have been resolved by the rebuttal resulting in 2 weak accept and 1 strong accept recommendation by the reviewers. A fourth reviewer who gave a weak reject did not give a final rating after the rebuttal. Having taken a look at their criticisms myself and the authors rebuttal, I am satisfied in general by the authors' response.

---

### Decision · Program_Chairs · 2024-04-05

Accept (Oral)